# Mammalian Cell-Growth Monitoring Based on an Impedimetric Sensor and Image Processing within a Microfluidic Platform

**DOI:** 10.3390/s23073748

**Published:** 2023-04-05

**Authors:** Ivana Podunavac, Teodora Knežić, Mila Djisalov, Nejra Omerovic, Marko Radovic, Ljiljana Janjušević, Dimitrije Stefanovic, Marko Panic, Ivana Gadjanski, Vasa Radonic

**Affiliations:** BioSense Institute, University of Novi Sad, Dr. Zorana Đinđića 1, Novi Sad 21000, Serbia

**Keywords:** microbioreactor, microfluidics, impedimetric sensor, biomass, inkjet printing, PMMA, interdigitated electrodes, scaled-down

## Abstract

In recent years, advancements in microfluidic and sensor technologies have led to the development of new methods for monitoring cell growth both in macro- and micro-systems. In this paper, a microfluidic (MF) platform with a microbioreactor and integrated impedimetric sensor is proposed for cell growth monitoring during the cell cultivation process in a scaled-down simulator. The impedimetric sensor with an interdigitated electrode (IDE) design was realized with inkjet printing and integrated into the custom-made MF platform, i.e., the scaled-down simulator. The proposed method, which was integrated into a simple and rapid fabrication MF system, presents an excellent candidate for the scaled-down analyses of cell growths that can be of use in, e.g., optimization of the cultivated meat bioprocess. When applied to MRC-5 cells as a model of adherent mammalian cells, the proposed sensor was able to precisely detect all phases of cell growth (the lag, exponential, stationary, and dying phases) during a 96-h cultivation period with limited available nutrients. By combining the impedimetric approach with image processing, the platform enables the real-time monitoring of biomasses and advanced control of cell growth progress in microbioreactors and scaled-down simulator systems.

## 1. Introduction

The scaled-down strategy is primarily used for the study of complex systems where the size and complexity of the system are reduced while the important physical and chemical characteristics are maintained. Through the development of a scaled-down simulator, i.e., a scaled-down version of the large system, simulations of the system’s behavior under precisely controlled conditions can be performed in a cost-effective manner. This approach allows for the development and optimization of large-scale processes while enabling a deeper understanding of the underlying processes since parameters and variables are significantly reduced in the microenvironment. In addition, this approach reduces the need for large-scale equipment and the use of reagents and materials [1,2,3,4]. Using the scaled-down approach, especially microfluidic platforms and scaled-down simulators [5] integrated with various sensing technologies, can be very beneficial for different bioprocesses to solve scaled-up problems [6,7]. The integration of sensors within microfluidic platforms has led to the development of lab-on-a-chip [8,9] and organ-on-a-chip platforms [10,11] which can simulate complex biological systems in a controlled environment. These platforms have the potential to revolutionize the fields of medicine and biotechnology by allowing for the rapid and accurate testing of drugs and medical treatments, as well as providing insights into the functioning of biological systems [10,11,12].

The scaled-down approach is particularly relevant to the study of cultivated meat production, which is viewed as a viable alternative to traditional animal-based meat production due to its potential for a more ethical and sustainable bioprocess based on cell cultivation and tissue engineering techniques. Cultivated meat potentially offers solutions for major problems in traditional meat production methods [13]. Considering that traditional agriculture has huge impacts on pollution [14] and ongoing climate change [15] while at the same time facing challenges with the application of antibiotics and fungicides in animal production systems [16], the production of cultivated (cultured/lab-grown) meat has a huge potential to mitigate all the mentioned problems related to conventional meat production. The large-scale production of meat is especially important since the United Nations predicted that population growth will reach 10 billion people in the next 30 years, which will result in an increased food demand of more than 70% [17]. However, the production process of cultivated meat is still very expensive, and therefore, decreasing the costs of its production is one of the main priorities and attractive fields of research. In this sense, sensors can be particularly helpful at the production scale since they allow for improved in-process control and the re-optimization of the culturing process, saving on cultivation medium usage and lowering the overall price of the bioprocess. Different sensing solutions have been proposed in the literature for monitoring the relevant parameters of cell growth in a bioreactor. They are summarized in our previously published review articles [18,19].

Biomass is one of the most important parameters to monitor during the cultivation of cells in bioreactors since it directly describes the progress of cell growth. Many non-invasive techniques based on cell counting with hemocytometers [20], near-infrared spectroscopy [21], and dielectric spectrometry [22] have been proposed for biomass monitoring. Besides the previously mentioned techniques, which are considered as direct monitoring techniques, indirect methods have been proposed based on measurements of glucose and lactate concentrations [23], released gases during the bioprocess [24], and changes in the redox potential [25]. All of the proposed solutions have their drawbacks, and their integration into bioreactors is still a challenging task for commercial applications.

The impedimetric principle of cell growth detection has shown promising potential for biomass monitoring by using alternating current (AC) potential in the radio-frequency range [3,26], namely, the dielectric properties of cells strongly depend on the frequency of the applied field since the cells are heterogenous structures made of isolating membranes and conductive cytoplasm surrounded by a conductive medium. In other words, cells behave like small capacitors, and thus, their capacitive properties can be used for monitoring their growth over time. In addition, information about the resistive properties of the culturing system (which comprises cells with a culturing medium) and the impedance, as a complex quantity, also contains information about the capacitive properties of the culturing system through real and imaginary parts, respectively. This impedance information is then utilized to calculate characteristics such as cell viability, proliferation, and adhesion [27,28,29,30,31]. Furthermore, as a non-invasive technique, impedance sensing enables detection without any labeling method. Recent studies have shown that this approach is efficient both in 2D and 3D cell culturing systems, i.e., for both adherent cells in monolayers and free-floating cells in suspensions [32]. Consequently, the impedimetric approach has become a valuable method in numerous biological fields, including drug testing, the examination of cell movement, and tissue engineering [28,33,34,35].

In this paper, we present a microbioreactor platform with an integrated impedimetric sensor for monitoring mammalian cell growth over time. A multilayer structure that combines IDE configuration and aF MF chamber is proposed, providing both cultivating and sensing platforms for cell growth in the proposed chip. The MF system/scaled-down simulator was realized by combining cost-effective materials and the rapid fabrication technologies of inkjet printing and laser micromachining. The sensing potential was validated during 96 h of MRC-5 cell line cultivation inside the MF system, and additional information on the biomass was gained through the image processing of microscope images during the cultivation time. The proposed solution has shown excellent potential for monitoring the cell growth in a microbioreactor, particularly in terms of fabrication complexity, cost, and sensitivity.

## 2. Microfluidic Platform Layout

The layout of the proposed MF platform is presented in Figure 1. The platform comprises a microbioreactor and impedimetric sensor integrated into a compact, multifunctional chip/scaled-down simulator that consists of different layers realized using laser micromachining. The glass layer (Layer 5) was used as a supporting layer for the inkjet-printed impedimetric sensor. The proposed sensor’s design has an interdigitated capacitor form, and its structure was optimized with inkjet printing parameters in order to obtain a good resolution of the printed layer. In previous studies, SU-8 resist was widely used in different biomedical applications due to its good biocompatibility [36,37,38]. Based on this, an additional layer of the SU-8 resist was used to cover the IDE and prevent direct contact between the electrode from the cells and the medium. The chamber for the cell cultivation has a volume of 1.5 mL and it was realized in the middle layer (Layer 3), which contains inlet and outlet holes and a chamber with a rounded-edge design to prevent the formation of bubbles inside the chip. The chamber is closed from the top (Layer 1) and bottom side (Layer 5), where the holes for the inlet and outlet are aligned with Layer 1. The impedimetric sensor is aligned to fit the chamber, while the connection to the sensor is provided with connecting fingers that are outside of the chamber. The connection between layers is enabled by two interconnecting Layers, 2 and 4, which were realized in the same design as that of Layer 3.

## 3. Theoretical Background

Biological structures are heterogeneous and exhibit both conductive and dielectric properties in an applied AC field. The behaviour of these systems is described by quantities such as dielectric permittivity and conductivity, which are functions of the applied frequency and have dispersion characteristics. In heterogeneous materials such as biological structures, different polarization mechanisms and dielectric relaxation at specific frequencies lead to the appearance of four areas (*α*, *β*, *γ*, and *δ*) in the frequency spectrum [31,39]. The proposed impedimetric approach for cell growth monitoring is based on dielectric permittivity sensing, which highly depends on the frequency range used for measurements. The cell membrane has capacitive properties and a high resistivity for direct current (DC) and low frequency alternating current. Therefore, radio frequencies up to 100 kHz are suitable for biomass sensing since charges accumulate on the membrane surface and the cells behave like dipoles. The system can be observed as a collection of spherical capacitors, and the capacitance of the system increases with an increasing number of cells. The impedimetric approach is suitable for monitoring cell growth as only living cells have the previously described properties due to their intact cell membrane, which enables charge separation. The *β*-dispersion region exhibits low conductivity in biological structures. An impedimetric sensor has been used in the controlled environment of a microbioreactor platform for cell growth monitoring [26,40,41].

As the proposed sensor is covered with a resist, it measures the capacitive properties of the system, and therefore, the imaginary part is dominant in the overall impedance and reflects the number of cells in the system, while the real part is affected by the electrical conductivity of the medium and cells. The proposed system with the integrated sensor can be observed as a collection of spherical capacitors, as shown in Figure 2a,b [42,43], where the capacitance of each layer can be determined based on their dielectric properties and distances from the sensor surface. The total capacitance is dependent on the properties of the materials used (glass and SU-8), as well as the cells and medium. Since all other parameters in the system are constant, the imaginary part of the impedance is determined by the concentration of cells. Based on this, the imaginary part of the impedance, which is inversely proportional to the total capacitance, will decrease with the increase in cell concentration. On the other hand, the real part of the impedance, which is related to the electrical conductivity of the medium surrounding the sensor, can provide information about the changes in the medium surrounding the sensor, such as the presence of ions or changes in the ionic strength of the solution.

The total impedance can be modelled with an equivalent electronic circuit, as shown in Figure 2c. The proposed model contains elements that describe the resistance of the cell medium R_med_, the resistance of and capacitance of the cells, R_cell_ and C_cell_, and the resistance and capacitance of the other components comprise the microfluidic system, R_el_ and C_el_, respectively. The parallel resistor-capacitor (R_cell_C_cell_) circuit for cells describes the effective contribution of the formed cell monolayer without considering a cell-specific contribution. In addition, the R_el_C_el_ circuit of electrodes contains an inkjet-printed sensor and an additional thin-film layer that prevents direct contact between the cells, substrate, and PMMA.

## 4. Materials and Methods

### 4.1. Materials

#### 4.1.1. Cell Line

The cell line MRC-5 (human fibroblasts, ATCC CCL-171) was used in this study as a model of healthy mammalian adherent cells. Before being seeded into the microbioreactor platform, the MRC-5 cells were grown in standard tissue flasks (Costar, 25 cm^2^) in Dulbecco’s modified Eagle’s medium (DMEM, Sigma-Aldrich, St. Louis, MO, USA) with 4500 mg/L of glucose, supplemented with 10% fetal calf serum (FCS) (Sigma-Aldrich) and a 1% antibiotic/antimycotic solution (Sigma-Aldrich). The cells were seeded in a concentration of 0.7 × 10^6^ cells, incubated at 37 °C in a 100% humidity atmosphere with 5% of CO_2_ in a CO_2_ incubator (ICO50, MEMMERT, Schwabach, Germany), and passaged twice per week. For seeding the cells into the microbioreactor, the cells in t25 flasks were trypsinized using 0.25% trypsin-EDTA (Serva, Heidelberg, Germany), counted, and seeded at a concentration of 50,000 cells/mL. DMEM with glucose, FCS, and antibiotic/antimycotic was used for the microbiorector cultivation, as well.

#### 4.1.2. Microfluidic Platform Fabrication

For the fabrication of the multilayered MF platform, transparent and biocompatible materials were used (glass and poly(methyl methacrylate) (PMMA) (Oracal Polikarbonati, Novi Sad, Serbia). The impedance sensor was realized in the form of an IDE and printed on a 2 mm-thick glass slide. The IDE structure was printed with a commercial nano silver ink with 15% of Ag nanoparticle-loading (Agfa-Gevaert N.V., Mortsel, Belgium). The sensor and the glass layer were covered with an 8 µm-thick SU-8 3000 resist (Micro Chem, Durham, UK). The top and the middle layers were manufactured in 2 mm-thick PMMA, and the interconnecting layers were made using 3M double-sided adhesive tapes (3M™ GPT-020F, St. Paul, MN 55144-1000, Minneapolis, MN, USA).

### 4.2. Methods

#### 4.2.1. Sensor Manufacturing

The electrodes for the impedimetric sensor were fabricated using inkjet printing technology, using a piezo-controlled inkjet printer (Fuji Dimatix DMP-3000, Tokyo, Japan). The printing was performed in two layers on the glass substrate with the drop spacing set at 25 microns. The distance between the two adjacent fingers of the IDE structure was 630 microns and the thickness of the individual finger was 350 microns. After printing, the structure was annealed at 200 °C for 30 min. Characterization of the IDE design was completed with a profilometer (Huvitz Panasis, Gunpo, South Korea) and scanning electron microscopy (SEM) (Hitachi TM3030, Ibaraki, Japan).

In order to avoid direct contact between the cells and the sensor surface, a thin layer of the biocompatible SU-8 photoresist was deposited with the doctor blade technique on the sensor’s surface. The deposited thin layer was pre-baked on a hotplate (Hot Plate Magnetic Stirrer–IKA RCT Basic, Guangzhou, China) for 10 min at 65 °C. The used photoresist SU-8 3000 was a negative resist, and so the thin film was directly exposed to UV light with a screen copy unit (Technigraf Variocop S, Grävenwiesbach-Hundstadt, Germany). The deposited layer was made of SU-8 resist and cyclopentanone mixed at a mass ratio of 1:5. After the deposition to the sensor’s surface, the post-bake was performed for 30 min at 95 °C.

The top and middle layer designs were precisely cut with a CO_2_ laser (CNC—MBL 4040RS, Minoan Binding Laminating, Belgrade, Serbia) in PMMA. Afterwards, all layers were bonded in a cold lamination process with 3M double-sided adhesive tapes. Finally, the connection between the sensor and the electronic read-out was realized by fixing the twisted pair wires with the conductive paste (RS PRO silver conductive adhesive epoxy, Northants, UK).

#### 4.2.2. Impedance Measurements

Monitoring of the cell growth was completed over 96 h with limited nutrients available for the cell growth, i.e., without changing the medium inside the microbioreactor. In this way, after consuming all the nutrients during cell growth, the dying phase was enabled to be detected as the final phase of the cell growth process. The impedance measurements were completed with the potentiostat/galvanostat/impedance analyzer PalmSens4 (PalmSens BV, Houten, The Netherlands) and PSTrace 5.8 software, module Electrochemical Impedance Spectroscopy in the frequency range 5–100 kHz, with an amplitude value of 0.01 V. In order to discuss the electrical properties of the cells at different frequencies, the impedance results have been presented at 10, 50, and 90 kHz.

#### 4.2.3. Equivalent Circuit Modeling

The impedance results were modelled with an equivalent electronic circuit using an open-source EIS spectrum analyzer [44]. The proposed model contains quantities that describe the resistance of the cell medium R_med_, the resistance of and capacitance of the cells, R_cell_ and C_cell_, respectively, and the resistance and capacitance of the electrodes, R_el_ and C_el_, respectively. The parallel resistor-capacitor (RC) circuit for the cells describes the effective contribution of the formed cell monolayer without considering a cell-specific contribution. In addition, the RC circuit of the electrodes contains an inkjet-printed sensor and an additional thin-film layer that prevents direct contact between the cells and the electrodes.

#### 4.2.4. Image Processing

In order to estimate the cell concentration inside the microbioreactor, the surface coverage (SC) of the cells was investigated with an image processing technique using microscope images (Nikon ECLIPSE Ts2R). A control chip, fabricated and seeded using all the steps of the described procedure, was used for imaging during the cultivation time of 96 h.

To extract information about the cell coverage on the chip, parts of microscopic images that were covered with cells were segmented. The proposed image-processing procedure for segmenting cells consists of 3 key steps: pre-processing, edge detection, and the post-processing step.

Pre-processing step:

The microscopic images were first converted from RGB to grayscale format. Since micro-impurities appear during imaging and acquisition sensor imperfections affect image quality, some sort of filtering has to be applied in a way that will reduce unwanted effects and preserve important features of cells. Therefore, a median filter [45] was used, which is a non-linear filter for noise removal that determines new pixel values by calculating the median across a predefined neighborhood kernel. These kernels are usually *n* × *n* matrices surrounding the pixel that needs to be filtered, where *n* is a predefined dimension. In this study, the kernel size was set to *n* = 7. This type of filter is often used as a pre-processing step for edge detection algorithms as it suppresses the noise but preserves the edges in an image. Afterward, histogram equalization was applied to the over-filtered images using Contrast Limited Adaptive Histogram Equalization (CLAHE) [46], which corrects contrast and makes objects of interest more distinguishable with respect to the background.

Edge detection step:

The main part of the proposed cell segmentation procedure utilizes a Canny edge detector [47], which is used to extract edges from preprocessed images. This method consists of several steps: filtering, gradient calculation, non-maximum suppression, double thresholding, and edge tracking. First, images are filtered with a smoothing Gaussian filter, and then, using the Sobel operator [48], the intensity gradient is calculated. Then, non-maximum suppression thins the edges by suppressing pixels that are not at the maximum edge strength in the direction of the gradient. Double-thresholding or Hysteresis thresholding decides which edges are strong and which are not. Two threshold values are defined: *T_max_* and *T_min_*. Edges with an intensity gradient greater than *T_max_* are declared as edges, and those below *T_min_* are declared as non-edges and are thus discarded. Edges that lie between these two thresholds are classified as edges or non-edges based on their connectivity. If they are connected to pixels declared as strong edges, they are considered to be part of the edges. Otherwise, they are also discarded. In this way, the Canny edge detector achieves thinner and more precise edges.

Post-processing step:

Mathematical morphology has been used as the final step of the image processing procedure. Generally, by constructing a special structural element/kernel such as a disk or cross, it is possible to manipulate and influence the final form of detected edges. A dilation operation adds pixels to the boundary pixels by the structural element while erosion shrinks boundary pixels in the same manner [49]. A segmentation of the microscope images was achieved using multiple dilations and erosions, respectively. The results in the form of a coverage percentage on the bioreactor have been extracted by dividing the number of segmented cell pixels and the total number of pixels in microscopic images, multiplied by 100. For each imaging, three images at the same magnification were taken at three randomly chosen places in the microbioreactor, and the means and standard deviations were calculated per stage.

Based on the known initial concentration of cells ($c_{i}$) and SC percentage for the initial concentration calculated from the imaging process ($SC_{i}$), an estimation of the cell concentration was completed for the microscope images during the 96 h of cultivation $\left(c_{t}\right)$. The increase in the cell concentration is directly proportional to the SC percentage in the microbioreactor over the cultivation time $\left(SC_{t}\right)$, which was calculated using Equation (1):(1)$$c_{t}=c_{i}\frac{SC_{t}}{SC_{i}}$$

## 5. Results and Discussion

### 5.1. Sensor Characterization

Figure 3a presents an inkjet-printed impedimetric sensor with an IDE design before integration into the microfluidic platform. Characterization of the realized IDE design was completed with a profilometer and SEM, as seen in Figure 3b,c. With its advanced features, inkjet printing is capable of producing sharp and smooth edges, resulting in detailed and accurate prints such as those seen in the profile image in Figure 3b. On the other hand, SEM images provide additional information about the defects, cracks, and other anomalies at the microscale that may affect a sensor’s performance. In Figure 3c, besides its composition and grain structure, which was made of Ag nanoparticles, the SEM image showed that the sensor did not have any damage at the surface, and it had the uniform thickness of the inkjet-printed layer (Figure 3c.) The realized sensor was covered with SU-8 resist, using the previously described procedure, in order to prevent direct contact between the cells and the electrode surface. Afterwards, the sensor was cleaned with isopropanol before integration into the MF platform.

A PalmSens potentiostat was used for the impedimetric measurements and placed in the CO_2_ incubator during cultivation. A Bluetooth connection between the PC and the PalmSens enabled cultivation inside the CO_2_ incubator without taking out the microbioreactors with the cells. This step was crucial for preventing contamination of the cells during the cultivation time (Figure 3d).

### 5.2. Impedance Results

Figure 4 presents the measured results of the impedimetric sensor during the 96 h at three different frequencies in the measured range of 5–100 kHz. During the 96 h of cultivation, the entire cell growth curve could be noticed. The steep drop in the impedance module and imaginary part could be noticed during the first 2 h due to the falling and attaching of the cells on the sensor’s surface. Further analysis depended on the frequency that was followed, namely, according to the dispersion curve, and in the *α* phase, the tangential flow of ions occurred across the cell surface in the range of 10 Hz–10 kHz frequencies. The results, as shown in Figure 4a, for 10 kHz showed that the sensor could observe cell growth without being able to detect the lag phase. Further, the stationary phase and dying phase were hardly seen. On the other hand, higher frequencies corresponded to the *β* phase in the dispersion diagram, where the accumulation of the charges on the cell membrane occurred. This property enables observing cells inside the chip and following their growth over time. Figure 4b,c shows the measured results at 50 kHz and 90 kHz during 96 h of cultivation, respectively. The sensor on both frequencies could detect the following different phases of cell growth: attaching on the sensor surface, the lag phase, the exponential phase, the stationary phase, and the dying phase. The results, however, demonstrated that the change in the impedance signal was greater at 50 kHz compared to 90 kHz, indicating that cell permittivity rises slightly at frequencies up to 100 kHz, as illustrated previously in the literature [41]. Finally, due to the limited amount of nutrients available for cell growth, after 60 h of cultivation, the dying phase began. Dead cells detached from the surface during dying phase, resulting in a slight increase in impedance. At the end of dying phase, the cells broke down, resulting in small decrease in impedance.

After 96 h, the control tests of the fresh and used medium were performed. The results, as seen in Figure 4d, showed that small differences existed in the range of 5–20 kHz between the real parts of the impedance of the fresh and used medium. The real part, as seen in Figure 4d, showed higher values for the used medium in comparison with the fresh medium, namely, during the cultivation process, the cells were using nutrients from the medium (which were electrolytes), and we supposed that the decrease in nutrient concentration over the cultivation time led to the decrease in the conductivity properties of the used medium, i.e., an increase in the real part of the impedance of the used medium. On the other hand, the imaginary part did not show any significant difference. In that sense, we hypothesized that the contribution to the sensors’ response originated from the cells and not from the changing medium properties over time, especially at higher frequencies, which had shown better sensing performances.

### 5.3. Image Processing Results

Table 1 contains the mean values and standard deviations of the SC percentages during the cultivation time. In addition, the image processing results were used for estimating the number of cells inside the microbioreactor based on the known initial concentration of cells. The low standard deviations of the SC percentages indicated that the cells formed a uniform monolayer over the surface of microbioreactor since results were calculated from three images. In addition, the SC percentages showed an increasing trend that continued during the lag, exponential, and stationary phases, after which the image processing no longer matched the impedimetric measurements. An additional explanation is given in Table 2, which contains the segmented images, where a comparison is shown between the raw microscope image and the segmented background and foreground (SC) images, respectively; namely, as a consequence of the limited nutrients available for growth, after 60 h of cultivation, the cells entered the dying phase. After this point, image processing becomes less effective, most likely because dead cells detach from the surface and, presumably, the cell breakdown process begins. In this phase, the cell membranes begin to degrade and cells lose their structural integrity and function, leading to cell death. In the end, only remnants of cell material could be seen in the microscope images after 72, 84, and 90 h of cultivation, leading to false SC detection.

### 5.4. Comparison between Modeling, Imaging, and Measured Results

The proposed microfluidic system with an integrated impedimetric sensor was modeled using an equivalent electronic circuit. The modeling step can be useful in the prediction and optimization of a sensor’s performances. The equivalent system had three components: the cell medium, the cells attached to the sensor surface, and the sensor covered with a protective layer. Therefore, the model was divided into three parts to describe each component separately. The equivalent circuit used for modeling is presented in the inset of Figure 5a. A cell medium is typically an aqueous solution containing ions, and as such, it is conductive. When modeling the electrical behavior of cells and their surroundings, the cell medium is simplified as a resistor because it provides a pathway for electrical currents between the cells and the electrodes [26]. Thus, the cell medium was modeled using a resistor (R_med_), while the cells and the electrode covered with a protective layer required the consideration of capacitive properties in addition to conductive properties. Considering that the applied voltage was equal across the resistor and the capacitor for the same component, a parallel RC circuit was used to describe the cells and the electrode. Finally, two parallel RC circuits were then connected in a series with R_med_.

The model was compared with the measured results of the impedance module at the initial moment and after 24, 48, and 96 h of cultivation. Figure 5a shows that the proposed model provided a good match with the measured results. Furthermore, in the proposed model, the parameters related to the electrode (R_el_ and C_el_) were considered constant since the electrodes did not interfere with the measured system and their properties did not change during the cultivation period. The key parameter for describing cell growth over time is C_cell_, which directly reflects the growth of cells in a system, as shown in Figure 5b. This parameter also shows all the phases of cell growth, comprising the lag, exponential, stationary, and dying phases.

The comparison of the image processing results and the modeling results in Figure 5b shows that both methods produced similar trends for the first 60 h of the cultivation period, i.e., until the dying phase began. Afterwards, the impedance model corresponded to the actual state, while results of the image processing did not show a reliable response.

Finally, both solutions can be applied independently in the continuous monitoring of cell growth in microfluidic platforms. The image processing method can be used for the rough estimation of a biomass. However, in the case of larger bioreactors, the application of this method is limited due to the impracticality of obtaining images without additional manipulation and sampling. On the other hand, the proposed sensor solution can be used to detect all phases of cell growth with great accuracy. The proposed impedance measurement concept can be adapted, and the proposed sensor can be redesigned for integration in different commercial bioreactors for biomass monitoring.

## 6. Conclusions

In this paper, two methods for monitoring cell growth within a microbioreactor platform/scaled-down simulator were proposed: one based on image processing and another one based on impedance measurement with an integrated impedimetric sensor. The potential of the proposed methods was examined during the 96 h cultivation process of MRC-5 cells with limited nutrients. Image processing was applied to microscope images to monitor cell growth progress, showing effectiveness only for live cells. In contrast, the impedimetric sensor successfully monitored cell growth during the complete cultivation process, detecting all phases of the cell growth curve including the lag, exponential, stationary, and dying phases. The proposed sensor solution is both cost-effective and easy to produce using low-cost materials and technologies. Moreover, the impedance sensor showed potential for detecting the whole cell growth curve, making it a valuable tool for the continuous monitoring of cell cultivation in scaled-down simulators over time. Future research will be focused on the application of the proposed sensor for different cell line measurements in organ-on-chip experiments as well as on a scaled-up modification to the sensor’s design, with integration into different commercial bioreactors.

## Figures and Tables

**Figure 1 sensors-23-03748-f001:**
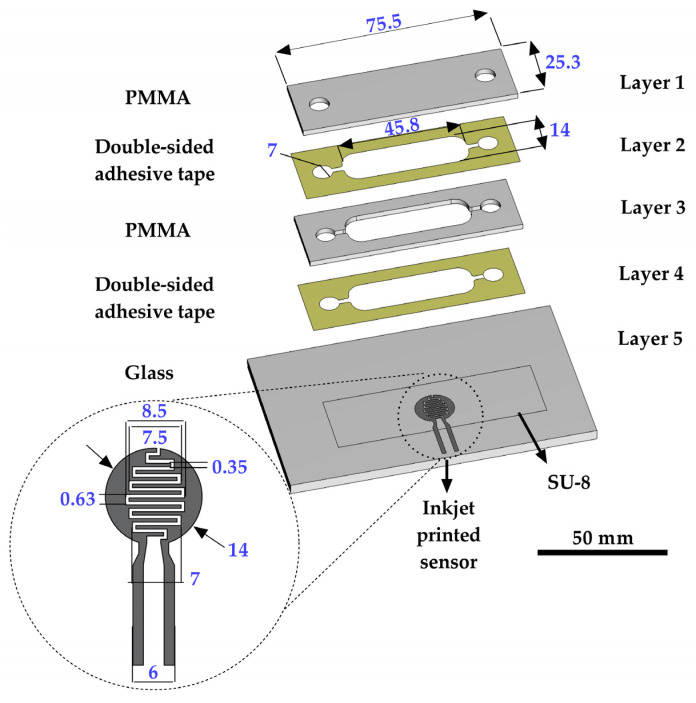
The multilayer structure and dimensions of the MF platform with an integrated inkjet-printed IDE sensor. All dimensions labeled in the figure are in mm.

**Figure 2 sensors-23-03748-f002:**
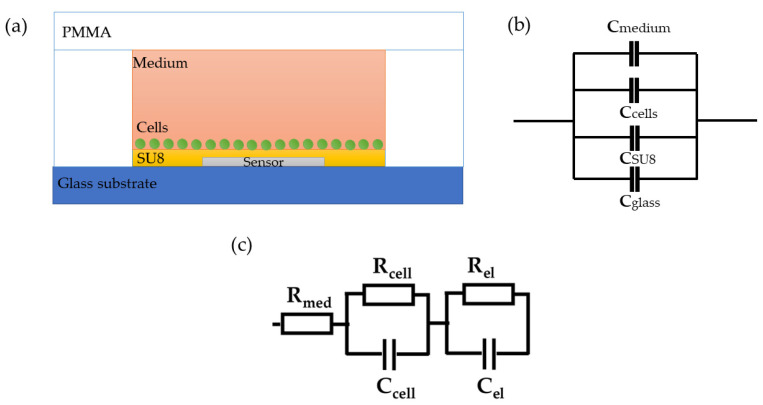
(**a**) Cross-section of the proposed sensor. (**b**) Measured equivalent capacitance. (**c**) Simplified electrical model.

**Figure 3 sensors-23-03748-f003:**
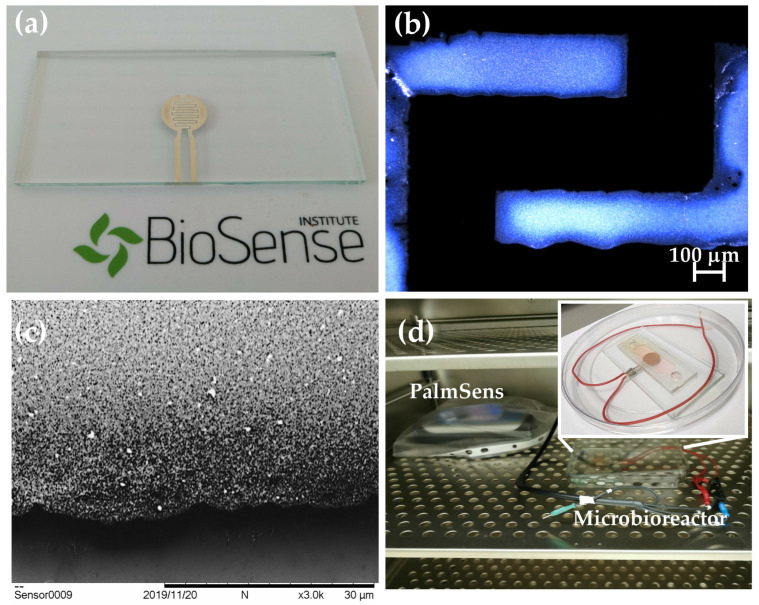
(**a**) Inkjet-printed impedimetric sensor on the glass substrate. (**b**) Profile image of the inkjet-printed IDE. (**c**) SEM image of the inkjet-printed IDE at a magnification of 3000×. (**d**) Experimental setup. A portable potentiostat PalmSens was placed inside of the CO_2_ incubator, together with a microbioreactor with an impedimetric sensor.

**Figure 4 sensors-23-03748-f004:**
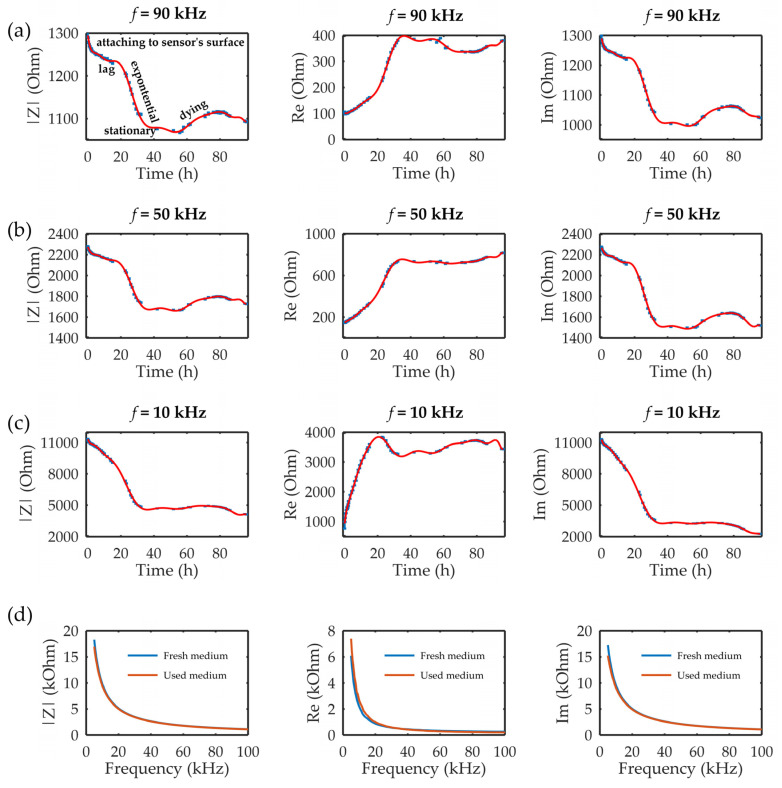
The impedance module (|Z|) and the real (Re) and imaginary (Im) parts of the impedance at three different frequencies in the measured frequency range: (**a**) 90, (**b**) 50, and (**c**) 10 kHz during 96 h of MRC-5 cultivation in the microbioreactor. The error bars in the graph represent standard deviations. (**d**) Control measurements for comparisons of fresh and used medium, i.e., medium after 96 h of cultivation, in terms of |Z| and the Re and Im parts.

**Figure 5 sensors-23-03748-f005:**
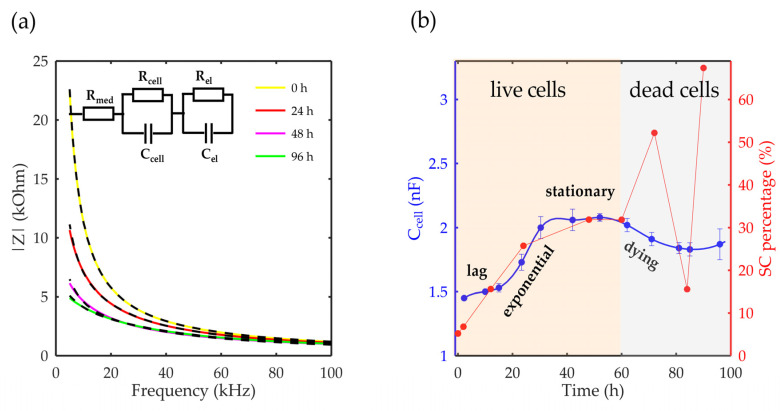
Fitting results: (**a**) The impedance module in the function of frequency for 0, 24, 48, and 96 h of cultivation. The black curves represent the fitted results, with the equivalent circuit presented in the inset. (**b**) Cell capacitance in the function of the cultivation time compared with the image processing results. The error bars in the graph represent the standard deviations of the equivalent circuit model and the image processing from images at three different places in the microbioreactor.

**Table 1 sensors-23-03748-t001:** Image processing results. The mean values and standard deviations of the surface coverage during the 96 h cultivation time are shown.

Time (h)	Mean Surface Coverage (%)	Standard Deviation	Estimated Cell Concentration (Cell/mL)
2	6.781	3 × 10^−3^	5 × 10^4^
12	15.592	5 × 10^−3^	1.2 × 10^5^
24	25.73	5 × 10^−2^	1.9 × 10^5^
36	25.64	7 × 10^−2^	1.9 × 10^5^
48	31.88	5 × 10^−2^	2.4 × 10^5^
60	31.86	5 × 10^−2^	2.4 × 10^5^
72 *	52.2	2 × 10^−1^	3.8 × 10^5^
84 *	15.6	1 × 10^−1^	1.1 × 10^5^
90 *	67.39	6 × 10^−2^	4.9 × 10^5^

* Grey rows—dying phase.

**Table 2 sensors-23-03748-t002:** Image processing procedures of cell detection: microscopic (left), segmented background (middle), and segmented foreground (right) images. The scale bar in the images is 0.1 mm, and the microscope magnification was 10×.

Time (h)	Microscopic Picture	Segmented Background	Segmented Foreground
0	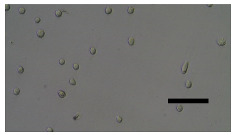	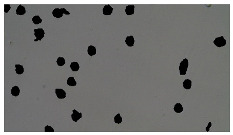	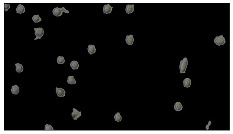
2	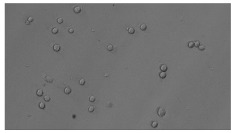	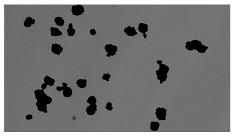	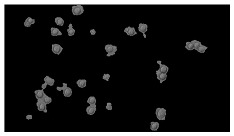
12	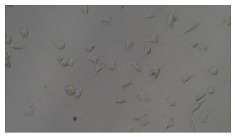	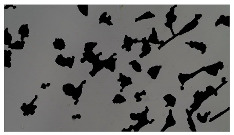	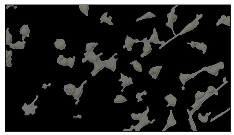
24	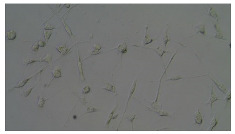	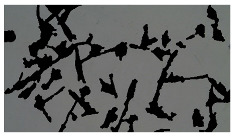	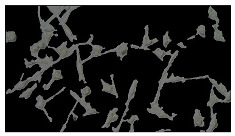
36	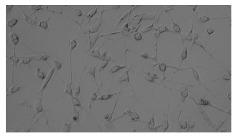	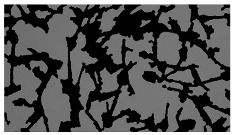	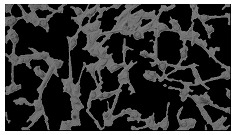
48	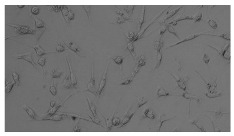	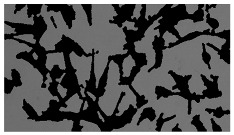	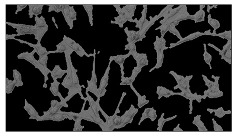
60	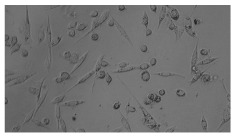	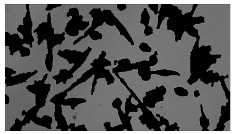	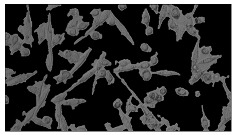
72	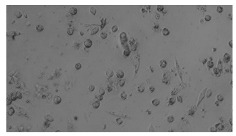	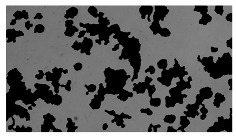	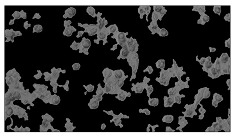
84	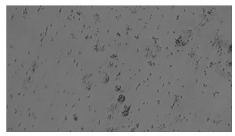	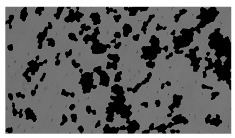	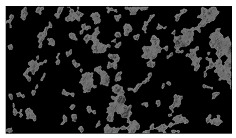
90	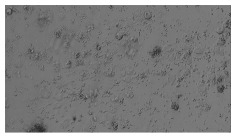	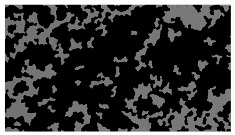	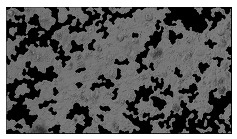

## Data Availability

The data presented in this study are available on request from the corresponding author.

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
