# Peer review of "Mammalian Cell-Growth Monitoring Based on an Impedimetric Sensor and Image Processing within a Microfluidic Platform"

_sensors, 2023, doi:10.3390/s23073748_

Round 1

Reviewer 1 Report

This paper proposed a novel method for monitoring cell growth by an impedimetric sensor on a microfluidic chip. The author detailly investigated the theoretical analysis and experimental applications. This holds great value for future research in the fields of organ-on-a-chip and cultivated meat. To ensure the advantages, understandability and logic of this paper, a major revision is required:

1)     What’s the material of the ink for the impedimetric electrodes? Is it toxic to the cells? The authors need to give the control group to analyze the cell viability on these materials.

2)     The authors indicated that after 60hs, the cells entered the dying phase. However, the authors need to show how to induce the cells into the dying phase.

3)     In the introduction, the authors illustrated the potential application of this technique is the monitoring of cultivated meat. So it is supposed to be that the cell concentration on the surface should be much high. In the demonstrated experiments, the cell density is not high enough, such as the cells at 60hr. The authors should test the results when the cell concentration is high enough.

4)     There are too many abbreviations that are not necessary, such as microfluidic (MF), cultivated meat(CM), etc. The author should reduce some.

Author Response

Dear Sir/Madam,

We appreciate the careful reviewing of our manuscript and we gratefully acknowledge the reviewers and editor for their useful comments. We have modified the manuscript to answer the questions raised by the reviewers. Our answers, some additional comments, and explanations are given in the text below and the changes are made in the resubmitted manuscript marked in red.

We thank you for the opportunity to improve our paper, including your comments and suggestions.

Sincerely,

The authors

Answers to the Reviewer’s Comments

Answers to the First Reviewer’s comments

This paper proposed a novel method for monitoring cell growth by an impedimetric sensor on a microfluidic chip. The author detailly investigated the theoretical analysis and experimental applications. This holds great value for future research in the fields of organ-on-a-chip and cultivated meat. To ensure the advantages, understandability and logic of this paper, a major revision is required:

Our reply: We appreciate the careful reviewing of our manuscript and we gratefully acknowledge the reviewer. Our answers and additional explanations are given below.

Comment 1: What’s the material of the ink for the impedimetric electrodes? Is it toxic to the cells? The authors need to give the control group to analyze the cell viability on these materials.

Our reply: We thank the reviewer for the valuable comment. The authors would like to point out that material of the ink is not in direct contact with cells and medium. The proposed inkjet-printed sensor is covered with the protective SU-8 layer which enables capacitive sensing of the cells seeded on top of the SU-8 layer. The description of sensor manufacturing is given in the paper in lines 189-190:

The IDE structure is printed with the commercial nano silver ink with 15% of Ag nanoparticles loading (Agfa-Gevaert N.V., Mortsel, Belgium). The sensor and the glass layer are covered with 8 µm-thick SU-8 3000 resist (Micro Chem, United Kingdom).

and lines 206-208:

In order to avoid direct contact between cells and the sensor surface, the thin layer of the biocompatible SU-8 photoresist has been deposited with the doctor blade technique on the sensor’s surface.

In previous studies, SU-8 resist has been used in different biomedical applications such as platforms for cell culture and cell encapsulation, immunosensing, neural probes, implantable pressure sensors, and adhesion layer, and its biocompatibility was confirmed in different cultivation studies [1–3]. Therefore, the authors added additional references in the resubmitted manuscript to provide support for SU-8 deposition on the inkjet-printed sensor.

  1.    Matarèse, B.F.E.; Feyen, P.L.C.; Falco, A.; Benfenati, F.; Lugli, P.; deMello, J.C. Use of SU8 as a stable and biocompatible adhesion layer for gold bioelectrodes. Sci. Rep. 2018, 8, 5560, doi:10.1038/s41598-018-21755-6.
  2.    Nemani, K.V.; Moodie, K.L.; Brennick, J.B.; Su, A.; Gimi, B. In vitro and in vivo evaluation of SU-8 biocompatibility. Mater. Sci. Eng. C Mater. Biol. Appl. 2013, 33, 4453–4459, doi:10.1016/j.msec.2013.07.001.
  3.    Chen, Z.; Lee, J.-B. Biocompatibility of SU-8 and Its Biomedical Device Applications. Micromachines (Basel) 2021, 12, doi:10.3390/mi12070794.

Comment 2: The authors indicated that after 60hs, the cells entered the dying phase. However, the authors need to show how to induce the cells into the dying phase.

Our reply: As mentioned in the paper, the dying phase of the cells in the proposed platform was induced by the limited nutrients available for cell growth and the small volume of medium (1.5 mL) in the microbioreactor environment. During the 96-hour cultivation period, the medium was not changed, and the cells were exposed to stressful conditions without nutrients, leading to the dying phase. Furthermore, inducing the dying phase of cells can be achieved through various methods, such as removing growth supplements from the cell medium or exposing cells to UV light. However, in this study, a new sensing solution based on an impedance sensor was proposed and we focused on inducing the dying phase of cells through the limited nutrients available in the microbioreactor environment [R1, R2]. In addition, we provided microscope images in the paper, to visualize the dying phase, where the detaching of cells from the sensor's surface can be seen, and the sensor's signal follows the change in impedance. Different mechanisms for inducing the cells into the dying phase will be examined in future experiments.

[R1] https://www.sigmaaldrich.com/RS/en/technical-documents/technical-article/cell-culture-and-cell-culture-analysis/mammalian-cell-culture/cell-culture-troubleshooting-cell-death

[R2] https://www.leica-microsystems.com/science-lab/how-to-do-a-proper-cell-culture-quick-check/

Comment 3: In the introduction, the authors illustrated the potential application of this technique is the monitoring of cultivated meat. So, it is supposed to be that the cell concentration on the surface should be much high. In the demonstrated experiments, the cell density is not high enough, such as the cells at 60hr. The authors should test the results when the cell concentration is high enough.

Our reply: The authors appreciate the reviewer's suggestions. The purpose of the presented study was to provide a proof-of-concept for cell growth monitoring via an impedimetric approach in a scale-down microbioreactor environment. The potential application of the proposed scale-down simulator for the monitoring of cultivated meat was mentioned in the introduction as one of the possible areas of application. The authors agree that the cell density was not high enough to resemble the high cell concentrations that should be present in larger scale cultivated meat bioprocessing. However, the purpose of the present study was to introduce and evaluate the proposed sensor solution for monitoring cell growth in a scale-down platform, rather than to specifically investigate the monitoring of cultivated meat bioprocess in bench-top or higher scale system. We mentioned the application in the field of cultured meat because such a small-scale platform is very good for testing and validation of different types of sensors that are being developed for cultured meat field.

On the other hand, the sensor is capable of detecting higher concentrations of cells more effectively than lower ones. Therefore, the decision to use a small concentration as the initial one was deliberate, as it allowed for a more accurate assessment of the sensor's sensitivity.

We acknowledge that the performance of the platform could be further tested and optimized with higher cell densities, and we will consider this for future work in scale-up analyses. At this point, a small volume of medium with high concentrations of cells in the range similar to those in cultivated meat bioprocessing in the proposed microfluidic platform and a limited volume of medium would cause more rapid death of cells.

However, the authors provide some additional experiments that can simulate the sensor response for higher concentrations of cells which is directly proportional to the dielectric constant. Namely, previous research has shown that the response of the sensor is mostly dependent on the capacitance of the layer close the electrode [R3, R4]. This is further confirmed by the results of the sensor in the first 2 hours of measurement, where the cell concentration remains unchanged but the signal decreases due to the cell attaching to the sensor’s surface.

[R3] Rui Igreja, C.J. Dias, Analytical evaluation of the interdigital electrodes capacitance for a multi-layered structure, Sensors and Actuators A: Physical, Volume 112, Issues 2–3, 2004, 291-301, ISSN 0924-4247, https://doi.org/10.1016/j.sna.2004.01.040.

[R4] Rui Igreja, C.J. Dias, Extension to the analytical model of the interdigital electrodes capacitance for a multi-layered structure, Sensors and Actuators A: Physical, Volume 172, Issue 2, 2011, 392-399, ISSN 0924-4247, https://doi.org/10.1016/j.sna.2011.09.033.

In the proposed microbioreactor system, SU-8 layer is a cover layer of electrodes, and a substrate layer for the cells that are growing in a monolayer. Considering that capacitive properties of the cells and medium are determining the sensor’s response, the proposed design can be modeled by the following scheme to determine the equivalent capacitance of the system, Figure R1. The total capacitance is dependent on the properties of the materials used (glass, SU-8) as well as the cells and medium.

As it was shown in the paper, the dielectric properties of the medium do not change during cultivation process. Thus, the total capacitance is primarily determined by the properties of the layer containing the cells, i.e., the number of cells present in the system. Finally, all of the capacitive properties previously described have an influence on imaginary part of impedance. From that reason, the sensor response can be connected with effective dielectric constant which is describing effective capacitance of the medium and cells, while changing during cultivation period mainly due to the cell growth.

Figure R1. Multilayer structure of the microbioreactor system with equivalent circuit of capacitors.

To simulate the change in the dielectric constant of the effective medium during the cultivation process, additional experiments were conducted. A mixture of medium and ethanol was prepared with varying concentrations, ranging from 100% medium (dielectric constant 80) to 40% medium mixed with ethanol (dielectric constant 24), to emulate the layer of cells and media. The effective dielectric constant is estimated based on the Kraszewski formula [R5], where presents the effective dielectric constant of the mixed solution,  and dielectric constants of medium and ethanol,  and volume ration of medium and ethanol, respectively, in the final solution.

[R5] Kraszewski, A. Prediction of the dielectric properties of two-phase mixtures. J. Microw. Power 1977, 12, 216–222.

Figure R2 presents the impedance module, real and imaginary components obtained for the estimated dielectric constant of a medium and ethanol mixture. The results indicate that the sensor is responsive to changes in the dielectric constant, as evidenced by the decrease in impedance with an increase in the dielectric constant. This trend aligns with the capacitive model previously described, whereby the contribution of individual cell dielectric constants to the total effective dielectric constant increases with the number of cells.

Figure R2. Ethanol-medium mix test for effective medium theory.

However, from the already performed experiments, an estimation of cell concentration is made based on the measurements taken during the exponential growth phase. We have correlated the sensor response with the increase in cell concentration. Based on this linear correlation of cell concentration and imaginary part of impedance, we can estimate the sensor response for higher cell concentrations from the calibration curve presented in Figure R3.

Figure R3. Calibration curves for imaginary part of impedance and estimated cell concentration in the microbioreactor.

Comment 4: There are too many abbreviations that are not necessary, such as microfluidic (MF), cultivated meat(CM), etc. The author should reduce some.

Our reply: The authors appreciate the reviewer's suggestion. The number of abbreviations is reduced in the resubmitted manuscript.

Reviewer 2 Report

The manuscript details the development of an impedance-based sensor, with imaging complements, to actively monitor cell growth from sampling or as a separate assay. 

The article gives the author a very detailed introduction and theory/background.  Although much appreciated by a someone not familiar with the detailed theory, the authors may want to provide a shorted theory section and removed the figure.  The introduction of the four regions is more that the authors needs to understand the paper.  Also, the graph does not clarify the relationship of impedance measures (all data collected) to the permittivity and conductivity on the graph in Figure 1.  The authors should consider just referring the readers to a review article or book for more information on frequency selection.   

Second, The microfluidic platform layout and microfluidic fabrication sections can be combined in the Materials and Methods section (you should keep the figure).

In Table 1, there is no need to go to 9 or more decimal points on the surface coverage or 6 or more on the standard deviation - you only have 1 or 2 digits on the cell concentration.  Consider reducing to reducing numbers for 2-3 digits for ease of reading.

In the conclusion, could the authors elaborate on next steps and give insight on application in cultivated meats or other fields?

Overall, this was an enjoyable paper to read and I see the merit in the approach.

Author Response

Dear Sir/Madam,

We appreciate the careful reviewing of our manuscript and we gratefully acknowledge the reviewers and editor for their useful comments. We have modified the manuscript to answer the questions raised by the reviewers. Our answers, some additional comments, and explanations are given in the text below and the changes are made in the resubmitted manuscript marked in red.

We thank you for the opportunity to improve our paper, including your comments and suggestions.

Sincerely,

The authors

Answers to the Reviewer’s Comments

The manuscript details the development of an impedance-based sensor, with imaging complements, to actively monitor cell growth from sampling or as a separate assay.

Our reply: The authors acknowledge the reviewer's detailed review of the paper. Responses and additional explanations are given in below.

Comment 1: The article gives the author a very detailed introduction and theory/background. Although much appreciated by a someone not familiar with the detailed theory, the authors may want to provide a shorted theory section and removed the figure. The introduction of the four regions is more that the authors needs to understand the paper. Also, the graph does not clarify the relationship of impedance measures (all data collected) to the permittivity and conductivity on the graph in Figure 1. The authors should consider just referring the readers to a review article or book for more information on frequency selection.

Our reply: We thank the reviewer for the valuable comment. In the resubmitted manuscript, the shorter version of the theoretical background is presented with references. Additionally, the authors provide an explanation based on impedance measurements. Namely, considering the capacitive properties of the sensor which is mainly sensing the dielectric properties of the cells and medium, a measurement range is chosen in the frequency range of β dispersion where cells behave like small capacitors, and the system can be observed through effective capacitance that originates from cells and the dielectric permittivity of the system.

Comment 2: Second, The microfluidic platform layout and microfluidic fabrication sections can be combined in the Materials and Methods section (you should keep the figure).

Our reply: The authors are grateful for the reviewer's comment. However, we have decided to keep the section Microfluidic platform layout without combining it with the Materials and Methods section. Our intention was to provide readers with detailed information on the research idea together with the layout of the proposed platform and then to describe materials and fabrication processes in detail. The authors believe that the text organized in this way provides a reader with a step-by-step introduction to the research study.

Comment 3: In Table 1, there is no need to go to 9 or more decimal points on the surface coverage or 6 or more on the standard deviation - you only have 1 or 2 digits on the cell concentration. Consider reducing to reducing numbers for 2-3 digits for ease of reading.

Our reply: The authors appreciate reviewer's valuable remark. We changed the results according to the reviewer’s suggestion.

Comment 4: In the conclusion, could the authors elaborate on next steps and give insight on application in cultivated meats or other fields?

Overall, this was an enjoyable paper to read and I see the merit in the approach.

Our reply: The authors are grateful for the reviewer's comments. Considering that the proposed system presents a proof-of-concept for impedance sensing in the microbioreactor environment, the following studies will be focused on scale-up approach, modification of the sensor's design and integration into different commercial bioreactors for biomass monitoring as weel as the application of the proposed sensor in measurment of different cell lines in the organ-on-chip experiments. Additional sentence is added in the conclusion of the paper.

Line 445-448

Future research will be focused on the application of the proposed sensor for different cell lines measurements in the organ-on-chip experiments as well as on scale-up modification of the sensor's design and integration into different commercial bioreactors.

Reviewer 3 Report

see attached document

Author Response

Dear Sir/Madam,

We appreciate the careful reviewing of our manuscript and we gratefully acknowledge the reviewers and editor for their useful comments. We have modified the manuscript to answer the questions raised by the reviewers. Our answers, some additional comments, and explanations are given in the text below and the changes are made in the resubmitted manuscript marked in red.

We thank you for the opportunity to improve our paper, including your comments and suggestions.

Sincerely,

The authors

Answers to the Reviewer’s Comments

The manuscript at hand presents a Microfluidic device capable of monitoring cell growth using an impedimetric sensor. The authors present impedance measurement results at three different frequencies, microscopic images and equivalent circuit diagram modeling results. Using the impedance measurement techniques, the authors can distinguish the different phases of cell attachment, growth and death. The publication is well written in terms of structure and speech.

Our reply: The authors acknowledge the reviewer's detailed review of the paper. Responses and additional explanations are given in below.

Comment 1: My major concern is the reproducibility of the results. In Figure 4 and 5, error bars are presented, however, it is not mentioned, where this error stems from or how it is calculated. Same for Table 2, where standard deviation is written but if I understand correctly, it is for 3 pictures. I wonder, how big would be the error for 3 individual experiments, i.e. biological triplicates. Apart from that, I only have minor comments that should be addressed before publication:

Our reply: The authors are grateful for the reviewer's remark. The error bars in Figures 3 and Figure 4 were included to present the standard deviation of the impedance measurements at a specific point of cultivation time. The error bar in Figure 3 is calculated from 8 measurements at each point of cultivation time over the frequency range of 5 kHz – 100 kHz, while Figure 4 contains error bars from fitting the results with equivalent circuit. On the other hand, the standard deviation presented in Table 2 is calculated for three microscope images from randomly selected positions in the microbioreactor. By increasing the number of measurements per point or the number of images, the error should be reduced.

The authors acknowledge the reviewer's concern regarding the reproducibility of the results. In general, the cell growth rate depends on the cell passage number, medium freshness, and overall health condition of the cells. Reproducibility of the results is ensured for detecting different phases of cell growth and change of the signal magnitude for all of the phases. However, since cell growth rate depends on many factors and cannot be controlled precisely in the same way in repeated experiments, the results of each experiment are slightly different.

Comment 2: Citations 1 and 2 are very specific examples. I would appreciate a review paper summarizing the topic.

Our reply: The authors agree with the reviewer's suggestion. Two new references are added in the introduction part:

  1.    Castiaux, A.D.; Spence, D.M.; Martin, R.S. Review of 3D Cell Culture with Analysis in Microfluidic Systems. Anal. Methods 2019, 11, 4220–4232, doi:10.1039/c9ay01328h.
  2.    Coluccio, M.L.; Perozziello, G.; Malara, N.; Parrotta, E.; Zhang, P.; Gentile, F.; Limongi, T.; Raj, P.M.; Cuda, G.; Candeloro, P.; Di Fabrizio, E. Microfluidic platforms for cell cultures and investigations. Microelectron. Eng. 2019, 208, 14–28, doi:10.1016/j.mee.2019.01.004.

Comment 3: Line 95 MRC5 cultivation is mentioned but it should at least be mentioned that this is a cell line. Further, why has this cell line been chosen?

Our reply: The authors agree with the reviewer and the sentence is modified in accordance with the reviewer’s suggestion. In addition, MRC-5 cell line was chosen as a model of healthy mammalian adherent cells (fibroblasts).

Lines 172-173:

The cell line MRC-5 (human fibroblasts, ATCC CCL-171) is  used in this study as a model of healthy mammalian adherent cells.

Comment 4: The theoretical background would benefit from a short explanation on impedance and especially how the real and imaginary part should change when cells are measured.

Our reply: The authors agree with the reviewer's suggestion. In the resubmitted manuscript, the theoretical background section has been changed.

Comment 5: Line 120-126 are two sentences that are nearly identical.

Our reply: The authors are grateful for the careful review of the paper. We modified the theoretical background.

Comment 6: The SU8 thickness should be mentioned in line 184 already.

Our reply: The authors agree with the reviewer’s suggestion. The change was made in the resubmitted manuscript.

Line 190-193

The sensor and the glass layer are covered with 8 µm-thick SU-8 3000 resist (Micro Chem, United Kingdom).

Comment 7: The electrical connection of the sensor should be mentioned in the Material&Methods section. What wires have been used? How were they connected to the device?

Our reply: The authors appreciate reviewer's remark. For the connection of sensor with potentiostat, twisted pair wires were used with silver conductive adhesive epoxy.

Additional information is added in the resubmitted manuscript:

Line 218-221

Finally, the connection between the sensor and the electronic read-out has been realized by fixing the twisted pair wires with the conductive paste (RS PRO silver conductive adhesive epoxy, Northants, United Kingdom).

Comment 8: A description of the incubator is missing in the Material&Methods section.

Our reply: The authors are grateful for the careful reviewing of the manuscript. Additional information is added in the resubmitted paper with the following description:

Line 177-179

The cells are seeded in concentration of 0.7·106 cells, incubated at 37 ൦C in the 100% humidity atmosphere and 5% of CO2 in CO2 incubator (ICO50, MEMMERT, Germany), and passaged twice a week.

Comment 9: Figure 3a: why does the ink appear golden if it is a silver ink?

Our reply: The authors acknowledge the reviewer’s question. After printing the colour of the ink is completely black, while after annealing as it was described in the paper, the ink has yellowish colour.

Comment 10: Figure 3c presents a SEM image. What additional information does the reader gain?

Our reply: The authors acknowledge the reviewer’s question. Additional information is added to the resubmitted manuscript.

Line 304-306

With its advanced features, inkjet printing is capable of producing sharp and smooth edges, resulting in detailed and accurate prints like it can be seen in the profile image, Figure 3b. On the other hand, the SEM images give additional information about the defects, cracks and other anomalies at the microscale that may affect the sensor's performance. In Figure 3c, besides its composition and grain structure made of Ag nanoparticles, the SEM image shows that the sensor does not have any damage at the surface and the uniform thickness of the inkjet-printed layer.

Comment 11: Line 322: figure 4c not 4a

Our reply: We acknowledge reviewer's remark. The change was made in the resubmitted manuscript.

Comment 12: Line 355: cell breakdown is not explained. What does that mean? Might be trivial for biologists but not for other scientists.

Our reply: The authors agree with the reviewer's suggestion. Additional explanation is added in the resubmitted manuscript.

Line 377-382

After this point, image processing becomes less effective, most likely because dead cells detach from the surface and presumably the cell breakdown process begins. In this phase the cell membranes begin to degrade and cells lose their structural integrity and function, leading to cell death. In the end, only remnants of cell material can be seen in the microscope images after 72, 84 and 90 h of cultivation, leading to false SC detection.

Comment 13: Table 1: Why so many digits? I would recommend cutting to two digits at the surface coverage. Further the standard deviation to maximum 3-4 digits. Standard deviation is for 3 images at different locations? How was the cell concentration estimated?

Our reply: The authors agree with the reviewer's suggestion. The number of digits is reduced in the resubmitted manuscript.

Comment 14: Table 2: Cells seem to move around even after the adhering phase. What is the reason for that?

Our reply: As we noted in the paper, the cells seem to adhere to the surface of the sensor after falling onto it during the seeding. We have taken images of the cells at three different locations over time, and we cannot say with certainty whether they move based on the images alone. However, we did observe a slight decrease in impedance during the lag phase of the sensor, which corresponds to the spreading of cells on the sensor surface. During the exponential phase and growth of the cells, they elongate, afterwards divide, and create new cells, which also affects micro-movements on the surface and the coverage of the sensor's area. In that sense, we can assume that cells have some micro-movements in the microbioreactor during the cultivation process, which does not affect the results of the measurements.

Round 2

Reviewer 1 Report

The authors have replied to all my concerns. This paper is ready to be published.